# Establishing a National Community of Practice for Newborn Screening Follow-Up

**DOI:** 10.3390/ijns7030049

**Published:** 2021-07-26

**Authors:** Erin Darby, John Thompson, Carol Johnson, Sikha Singh, Jelili Ojodu

**Affiliations:** 1Association of Public Health Laboratories, Silver Spring, MD 20910, USA; sikha.singh@aphl.org (S.S.); Jelili.Ojodu@aphl.org (J.O.); 2Newborn Screening Laboratory, Public Health Laboratories, Washington State Department of Health, Shoreline, WA 98155, USA; John.Thompson@DOH.WA.GOV; 3Stead Family Children’s Hospital, University of Iowa, Iowa City, IA 52242, USA; carol-johnson@uiowa.edu

**Keywords:** newborn screening, follow-up, short-term follow-up, technical assistance, community of practice

## Abstract

Newborn screening (NBS) follow-up programs in the United States are managed at the state level, leaving limited opportunities for collaboration across programs and coordinated resource sharing. The Newborn Screening Technical assistance and Evaluation Program (NewSTEPs), a program of the Association of Public Health Laboratories (APHL), has established a national community of practice for NBS follow-up by creating a network of follow-up staff and stakeholders through education and engagement opportunities. The activities of NewSTEPs in support of NBS follow-up have strengthened information dissemination, collaboration, data collection and technical assistance-driven mentorship across the national system.

## 1. Introduction

In the United States, newborn screening (NBS) is recognized as one of the most significant public health achievements of the 21st century as it identifies thousands of newborns at increased risk of certain heritable disorders each year [1]. A critical component to the success of newborn screening systems is follow-up, which aims to ensure that all newborns receive screening and that results are shared with the appropriate caregiver. Once a positive result is found, follow-up programs track whether confirmatory or diagnostic testing has been completed and that the newborn receives treatment if necessary.

Follow-up programs have a unique set of challenges, from care coordination to data collection to program management, and have benefitted from the establishment of a national community of practice. The Association of Public Health Laboratories (APHL), a membership organization representing Public Health Laboratories, has robust resources to strengthen effective laboratory systems and its Newborn Screening and Genetics program collaborates with the Centers for Disease Control and Prevention (CDC) and with the Health Resources and Services Administration (HRSA) to improve the quality of newborn screening test results at state public health laboratories. As state newborn screening programs continue the expansion of their testing capabilities to meet the increasing number of conditions added to the Recommended Uniform Screening Panel (RUSP), the need to integrate the follow-up community into the national dialogue happening at APHL became apparent [2]. Follow-up programs are managed at the state level which allows for the customization of practice based on state demographics, geographic characteristics, the access to clinical referral networks, disorders screened and call out algorithms. However, without involvement from national organizations coordinating outreach, state-based follow-up programs would have little to no opportunities to connect across state lines. The value of establishing, maintaining, and strengthening communities of practice to engage in collective learning around a shared domain of human behavior, in this case newborn screening follow-up, is a well-defined social theory of learning [3]. As such, and in the absence of national standards, follow-up programs need a community of practice to share ideas and resources, to offer support and opportunities for professional development, and to build strong community networks.

## 2. Materials and Methods

### 2.1. Establishing a Dedicated Short-Term Follow-Up Workgroup

A program of the APHL, the Newborn Screening Technical assistance and Evaluation Program (NewSTEPs) [4] funded by the Health Resources and Services Administration (HRSA) in 2012, serves as a national technical assistance resource center to US-based newborn screening programs. In 2013, NewSTEPs established a dedicated short-term follow-up workgroup to provide technical assistance and networking forums for NBS follow-up programs. The workgroup is comprised of representatives from state programs ensuring accurate and informed representation throughout the country and meets monthly via teleconference to identify technical assistance needs within the follow-up community and to develop and support quality improvement initiatives in follow-up. The NewSTEPs Short-Term Follow-up Workgroup Goals are as follows:Strengthen the newborn screening system by providing input, guidance, and technical assistance on follow-up in newborn screening.Offer a forum for communication in which follow-up staff from regional and state newborn screening follow-up programs can network and collaborate on quality improvement efforts.Identify needs and offer newborn screening programs technical assistance related to short-term follow-up.

### 2.2. Standardizing Data Collection

Experts from the newborn screening community established a panel of eight Quality Indicators to track quality practices within and across the United States newborn screening system. The indicators underwent iterative refinement through consensus building across the NBS community, and were captured, tracked and analyzed in the NewSTEPs Data Repository [5]. The Quality Indicators track pre-analytic, analytic and post-analytic processes, offering a harmonized set of metrics by which to inform data driven outcome assessments and support tracking of quality improvements. Of these eight Quality Indicators, five (Table 1) were useful in quantifying areas for improvement within newborn screening follow-up [6]:

Newborn screening programs provide Quality Indicator data to the NewSTEPs Data Repository on a voluntary basis and are required to sign a Memorandum of Understanding (MOU) with the APHL for data privacy and data security purposes. Newborn Screening programs can review and analyze their own state metrics and compare them with aggregate national and regional metrics using real-time data visualizations provided on the NewSTEPs website. Quality Indicators for follow-up can be utilized to provide longitudinal comparisons over time and offer a standardized way to compare quality practices across state programs. Additionally, the public-facing NewSTEPs State Profiles curate and maintain characteristics of national short-term follow-up programs, such as definitions, follow-up time period for inconclusive diagnosis, existence of long-term follow-up activities, operating hours, and contact information with the purpose of serving as a national centralized location where follow-up staff can seek information about peer programs.

### 2.3. National Webinars

Since 2013, the Short-Term Follow-up Workgroup has hosted 30 national webinars on topics ranging from “Using Infographics for Data and Parent Materials” to “Emergency Preparedness for Newborn Screening Programs” and “Reducing Time from Referral to Treatment.” Most of these webinars also featured a spotlight on individual state programs to encourage sharing successes and challenges with the national follow-up community. The webinars sometimes feature outside experts, including clinicians, midwives, parent advocates, and ethicists, who bring new knowledge to the community, and they often feature members of the follow-up community itself to share their experience and receive feedback. National webinars are scheduled roughly every quarter but were postponed in 2019 due to a focus on taskforce projects and development of the NBS FLEX Program. They were halted again in the second quarter of 2020 due to the COVID-19 pandemic to allow organizers and participants to focus on COVID-19 response.

### 2.4. National Meetings

NewSTEPs has hosted two in-person national meetings, bringing together follow-up personnel from around the country as well as several international programs. Invitations to these meetings are offered to follow-up program staff from 50 states, DC, Puerto Rico and Guam, with the APHL providing travel support for one representative from each U.S. program; however, the meetings are open to additional staff and other NBS stakeholders at their own expense. The national meetings were objective driven (Table 2) with invited speakers and content determined by a planning committee comprised of a subset of the NewSTEPs short-term follow-up workgroup members.

The APHL hosts an annual abstract-driven Newborn Screening Symposium open to the entire global NBS community, during which one conference track is reserved for follow-up. The NewSTEPs short-term follow-up workgroup hosts an evening mixer for attendees from the follow-up community as a way to engage in informal networking and information sharing after a full day of lectures. Recently hired follow-up staff are encouraged to attend and introduce themselves while more experienced follow-up staff offer mentorship. The mixer always features an interactive activity to allow attendees to discuss their experiences and share ideas. The social nature of the event helps strengthen relationships and build trust within the community.

### 2.5. Taskforces

At the conclusion of the 2018 Short-Term Follow-up National Meeting, five taskforces were established to address areas of need within the community: (1) succession planning, (2) molecular literacy, (3) continuity of operations planning, (4) new hires, and (5) long-term follow-up. The new hire, continuity of operations planning and succession planning taskforces met to determine goals and objectives but were ultimately absorbed by broader committees focused on similar goals within the APHL Newborn Screening and Genetics program. The long-term follow-up and molecular literacy taskforces continued to meet, establishing and executing on planned deliverables over the course of several teleconferences and Zoom meetings.

### 2.6. Long-Term Follow-Up Taskforce

Long-term follow-up is an essential component of the care coordination system that begins after an infant has been diagnosed with a condition screened for by NBS and may extend throughout the lifetime of the individual [7]. Following the 2018 NewSTEPs Short-Term Follow-Up National Meeting, 11 individuals representing nine states joined the NewSTEPs long-term follow-up taskforce as volunteer leaders. The represented states were at various stages of long-term follow-up implementation, from an established program to partial implementation to no plan to implement. One of the first goals of the taskforce was to assess the current state of expanded follow-up activities across NBS programs. Drawing on previous work by the Association of Maternal & Child Health Programs (AMCHP) [8] and the American College of Medical Genetics and Genomics (ACMG) [9], the taskforce established a working definition of long-term follow-up to guide development of a survey aimed toward determining the type of activities performed, funding sources, data collection methods, and program needs in the development and/or maintenance of long-term follow-up within state newborn screening programs [10]. All members of the taskforce contributed to the development of the survey, which contained 20 questions [11]. The APHL’s Quality Systems and Analytics program distributed the survey to 74 individuals identified as follow-up managers or coordinators representing 53 state and territorial NBS programs, including dried bloodspot, critical congenital heart disease, and hearing screening programs. The APHL encouraged survey recipients to work with their staff to submit one survey per program. The survey was open from 9 January to 19 February 2020, receiving 42 responses. Of those, 32 were complete and the 10 incomplete surveys were excluded from analysis.

### 2.7. Molecular Literacy Taskforce

Following the 2018 NewSTEPs Short-Term Follow-Up National Meeting, 10 individuals representing eight states, Children’s National Medical Center, and the Centers for Disease Control and Prevention (CDC) joined the NewSTEPs molecular literacy taskforce. The goal of the NewSTEPs molecular literacy taskforce was to develop appropriate resources for NBS follow-up staff interacting with, analyzing and reporting out molecular screening results. To accomplish this, the taskforce conducted a survey of follow-up staff across all 53 U.S. NBS programs to determine the educational background of and identify any gaps in understanding of molecular terms among follow-up personnel [12]. In October 2019, the APHL distributed the 11-question survey to 68 follow-up managers and coordinators. They were encouraged to forward the survey to all members of their staff in order to assess the molecular literacy level and needs of the follow-up community.

### 2.8. Workshops

Since 2013, NewSTEPs has offered an annual Tandem Mass Spectrometry (MS/MS) Workshop to follow-up staff and medical directors. Attendees are required to apply and each year the program accepts 8 to 13 participants, limiting the size to encourage networking, relationship building, and an interactive forum for learning. This intensive five-day course led by MS/MS expert Dr. David Millington reviews the principles of MS/MS, diagnostic patterns in results, cut-offs, biochemical pathways, diagnostic follow-up, and biochemical and clinical features of the metabolic disorders. Workshop participants learn interpretive skills and diagnostic follow-up of certain disorders detectable through MS/MS screening, including amino acid disorders, urea cycle disorders, fatty acid oxidation disorders, and organic acid disorders. Successful course performance is recognized with a certificate of course completion and continuing education units are available.

### 2.9. Mentorship

Learning from other NBS programs through in-person interactions and site visits enhances collaboration and improves program workflows and processes [13]. In July 2019, short-term follow-up workgroup co-chairs (Carol Johnson, Iowa and John Thompson, Washington) submitted a Quality Improvement proposal to NewSTEPs in which NBS follow-up programs would be paired to provide peer-to-peer technical assistance. Their proposed program would provide a level of engagement and customized technical assistance that cannot be reached through other means such as national meetings and webinars. In October 2020, NewSTEPs established the Follow-up Learning EXchange (FLEX) Program. The program was designed to promote interactions for follow-up staff by matching peers between NBS programs and facilitating travel for in-person learning and mentorship. On 23 October 2020, APHL distributed a survey to NBS follow-up managers and coordinators across the U.S. as well as attendees of a FLEX information session held during the 2020 APHL NBS Symposium. Recipients were highly encouraged to work with their staff as a team to identify areas where the program could benefit from assistance and areas where they could provide support to others. They were asked to identify their top 3 immediate areas of need and top 3 areas of expertise [14]. Both need and expertise were identified in the survey with the understanding that an important feature of a community of practice is reciprocity, or the understanding that contributing to the overall value of the community will benefit all members [15]. NewSTEPs posited that identifying both mentor and mentee programs would nurture relationship building and skill transfer. The volunteer mentors in the FLEX program would initially contribute to the knowledge of a single program, but interacting with other FLEX members over time would offer opportunities for mutual learning in an ongoing fashion.

## 3. Results

### 3.1. Information Sharing and Dissemination

In the 2018 NewSTEPs Short-Term Follow-Up National Meeting evaluation, 94% of participants agreed or strongly agreed that they would apply what they learned at the meeting in their work and 90% agreed or strongly agreed that they had learned a new skill that would support NBS short-term follow-up. Reaching a primary meeting objective, 81% agreed or strongly agreed that the meeting helped them identify solutions to improve NBS short-term follow-up. Appreciation for the opportunity to network and collaborate with other follow-up programs was a common theme in the open-ended meeting feedback.

In a 2018 survey of a NewSTEPs short-term follow-up webinar, 94% of participants stated they were very satisfied or somewhat satisfied with the webinars and 85% stated they were very likely or somewhat likely to use the information learned from the webinar in their daily work. Each webinar (Appendix A) encouraged active participation, with a diverse array of topics covered within newborn screening the follow-up realm.

### 3.2. National Resource Development

Based on the results of the NewSTEPs long-term follow-up taskforce survey, half of the 33 states surveyed performed at least some long-term follow-up activities, while 41% indicated no plans to implement a long-term follow-up program. A key take-away was that each state defines and conducts long-term follow-up differently. The primary concerns from states about expanding their follow-up program were the lack of funding, little support from leadership and the absence of national guidelines or standards. Many respondents were not optimistic about the possibility for expansion of long-term follow-up in their state due to these limitations.

The NewSTEPs molecular literacy survey found that 44% of respondents communicate genetic or molecular results to primary care providers or other medical personnel on a daily basis. Another 20% communicate molecular results weekly. As a result of this survey, the taskforce developed a list of vocabulary words relevant to molecular testing within follow-up. The terms selected were compared with existing terminology lists from the CLSI and definitions were altered to accommodate all follow-up personnel regardless of their background in genetics. The terminology list developed by the molecular literacy taskforce serves as a resource to follow-up staff to improve the understanding of molecular terms and help staff communicate molecular screening results to providers and other medical personnel [16]. The collaborative nature of this project allowed for the creation of a tool that would meet the needs of the broader follow-up community and contribution from all taskforce members was essential to creating a comprehensive list of relevant molecular terms.

### 3.3. Data Analysis

The NewSTEPs State Profiles collect definitions used within national short-term follow-up programs, along with a description of long-term follow-up activities, updating them on an annual basis [17]. As of June 2021, of the 53 programs contributing information (50 states, District of Columbia, Guam, and Puerto Rico), the majority (*n* = 36) defined short-term follow-up as occurring “until diagnosis is made or ruled out.” An additional 10 programs defined it as “until the infant is on treatment,” 2 programs defined it as “until diagnosis is made or ruled out and the infant is on treatment (if indicated),” 2 programs defined it as “until confirmatory testing is performed,” an additional 2 programs defined it as “until confirmatory testing is performed and the patient is referred to the corresponding specialist,” and 1 program defined it as “borderline results: until the diagnosis is made/ruled out or 3 months (whichever comes first); presumptive positive results: until the diagnosis is made/ruled out or one year (whichever comes first.)”.

A varying number of states, but not all, are currently entering data into the voluntary NewSTEPs Data Repository to quantify quality practices as defined by harmonized Quality Indicators [5]. Challenges to data entry include the time required to perform voluntary data entry as well as limitations that exist in establishing data exchange between the laboratory, follow-up and clinical components of the newborn screening system within individual state programs. NewSTEPs initiated a Continuous Quality Improvement (CQI) coaching and connecting program to encourage states to complete their data entry by pairing them with a NewSTEPs staff member to act as a CQI connector. Connectors meet quarterly with state programs by phone or teleconference to identify program needs, including supplies, disorder status, and staff training as well as to discuss the data repository in order to increase engagement. Despite limitations to data collection, even a small number of states providing data can elucidate trends at the state level by which newborn screening programs can track improvement over time as well as identify potential intervention points for quality improvement practices. Quality Indicator 4—the percent of infants that have no recorded final resolution with the newborn screening program—captures the metrics that may help improve programs’ ability to track and reduce the number of infants that become lost to follow-up. Specifically, Quality Indicator 4a tracks, on an annual basis, the percent of infants that have no recorded final resolution by 12 months of age with the newborn screening program following the receipt of an unacceptable dried blood spot specimen. Quality Indicator 4c tracks the percent of infants that have no recorded final resolution by 12 months of age with the state newborn screening program following an out-of-range result from a dried blood spot screen requiring a further clinical diagnostic workup by an appropriate medical professional. Tracking Quality Indicator 4a would allow a program to determine if additional education and outreach were required to ensure that repeat specimens are obtained and received by the laboratory for testing. Tracking Quality Indicator 4c would allow a program to examine the feedback loop between the risk assessment (newborn screening) and diagnostic (clinical follow-up) components of the public health newborn screening system.

Table 3 provides a summary of the median percent of infants reported by participating state programs (out of a total of 53 states and territories participating in NewSTEPs) for Quality Indicators 4a and 4c. The median changes across years, but NewSTEPs cannot definitively state that the changes are significant until the time that additional states enter data into the voluntary data repository. However, state specific comparisons across years have proved useful, with NewSTEPs offering all NBS programs technical assistance to review and evaluate program specific data with the opportunity to initiate continuous quality improvement activities. The state specific data are protected by an MOU and are not intended to be shared publicly without the express written approval of each state. An analysis of the complete aggregate set Quality Indicator data will be published by the APHL upon receipt of a more complete dataset.

### 3.4. Education and Training

Since its inception in 2013, 81 follow-up staff from 41 NBS programs have attended the NewSTEPs MS/MS Follow-Up Workshop focused on the interpretation of biochemical newborn screening results. From a 2018 survey of participants, 100% agreed or strongly agreed that the material presented in the workshop will help them perform their job better and 57% of participants chose to participate in the workshop to exchange ideas with colleagues. In addition to learning from their peers, participants were also exposed to a leading expert in the field. With an average of 20 applications each year since 2016, NewSTEPs decided to expand the workshop offerings up to two sessions per year. Planning for the fall 2020 session was halted by the global COVID-19 pandemic, but the program will continue to meet the demand when the CDC releases guidance indicating that in-person meetings are safe.

Participants of the NewSTEPs FLEX Program identified areas of need where they would like assistance from another program, and areas of expertise where they could assist other programs. Areas of need and/or expertise include, but are not limited to, organization and infrastructure, workforce, communication and education, follow-up workflows and processes, emergency preparedness, data analytics and reporting, health information technology and interoperability, and long-term follow-up. In response to the FLEX survey, 16 programs indicated an interest in participating in both a mentor and mentee role. For the pilot phase of the FLEX program, NewSTEPs selected eight programs (by level of need articulated) to receive mentorship. Mentor programs who had indicated an interest in assistance from another program will have an opportunity to receive mentorship in subsequent rounds on an ongoing basis. Despite recent travel restrictions across the country preventing in-person meetings, programs still face ongoing challenges and unmet needs that require mentorship and training to resolve. Due to the COVID-19 pandemic, the NewSTEPs FLEX program launched virtually until the time that in-person opportunities can safely resume. The program is intended to be dynamic and flexible with multiple rounds of assistance offered as needs are addressed and new challenges arise. Beginning in January 2021, the first cohort of pairings met virtually, and mentees reported satisfaction with the guidance from mentors. Additional states have requested to join the NewSTEPs FLEX program and the second round of pairings will begin in fall 2021.

## 4. Discussion

Through the wide variety of activities previously described, NewSTEPs has developed and strengthened a vast network of follow-up experts, workforce, and stakeholders. Engagement in these activities continues to grow and programs are eager for more opportunities to learn and connect with the larger follow-up community. As a connection hub for state NBS follow-up programs, NewSTEPs has established a national community of practice for NBS follow-up which serves as a catalyst for follow-up program staff to form relationships across state lines to exchange ideas and improve their skills. In evaluation surveys from recent meetings, webinars, trainings, and other NewSTEPs programs focused on follow-up, participants commonly cited peer-to-peer networking as the main benefit. In a follow-up community of practice, staff and stakeholders interact on an ongoing basis to deepen their knowledge and expertise in follow-up practices. This peer-to-peer learning is the basis of a community of practice as they share information, insight, and advice [13]. NewSTEPs alone does not house all of the knowledge and resources of follow-up practice, but by offering varying opportunities to interact throughout the year, NewSTEPs has fostered and created forums for the follow-up community to maintain institutional knowledge and adapt to the rapidly changing nature of the practice.

The addition of later onset disorders to the RUSP has demanded increased attention toward long-term follow-up of newborn screening results. The results of the NewSTEPs long-term follow-up survey revealed that many states face significant barriers to implementing a long-term follow-up program within their newborn screening system. The complexities and advanced clinical and family engagement required of a robust long-term follow-up program requires the newborn screening follow-up system as it currently exists to receive additional support and resources from a broader stakeholder community. The U.S. Department of Health and Human Services Advisory Committee on Heritable Disorders in Newborns and Children (ACHDNC) has been examining long-term follow-up through its Follow-Up and Treatment Workgroup and featured a panel on the same during its February 2021 meeting [18]. Despite national conversations occurring at the federal level, a lack of uniform national guidelines for the implementation of long-term follow-up limits the rate at which states can gain support, resources and guidance at the programmatic and legislative levels around the expansion of their follow-up programs. NewSTEPs, as a national convener, has the opportunity to continue outreach efforts and data collection to examine how best to support the newborn screening system as states expand their follow-up programs. Specifically, proposing a harmonized definition of long-term follow-up and providing resources, training and capacity building to attain the parameters articulated in the definition will be productive next steps. The APHL does not endeavor to provide a uniform definition, but rather to provide support to programs that autonomously define their own short-term and long-term follow-up parameters.

## 5. Conclusions

The APHL and NewSTEPs have developed programs to connect the NBS follow-up community, resulting in an expanded knowledge sharing that has strengthened follow-up programs nationwide. With its many technical assistance resources, NewSTEPs serves as central point of contact for an evolving follow-up community of practice that requires continual growth and development. Long-term follow-up is in its infancy in the majority of U.S. newborn screening programs but requires additional attention, as disorders with later onsets and more complex treatment algorithms are added to state screening panels necessitating that programs examine care coordination and data collection beyond diagnosis. Attention toward long-term follow-up from federal groups such as the federal advisory committee ACHDNC shows promise for the future of expanded follow-up in newborn screening. Continued efforts from the APHL and NewSTEPs, in collaboration with advocacy groups, clinical networks, and the follow-up community at-large, will be needed to improve existing follow-up practices and to develop national standards for expansion.

## Figures and Tables

**Table 1 IJNS-07-00049-t001:** NewSTEPs Quality Indicators relevant to newborn screening follow-up.

Quality Indicator 3	Percent of eligible newborns not receiving a newborn screen, reported by dried blood spot or point of care screen(s).
Quality Indicator 4	Percent of infants that have no recorded final resolution (confirmed diagnosis or diagnosis ruled out by an appropriate medical professional) with the newborn screening program.
Quality Indicator 6	Percent of infants with an out-of-range newborn screening result requiring clinical diagnostic workup reported by disorder category.
Quality Indicator 7	Percent of disorders detected by newborn screening with a confirmed diagnosis by an appropriate medical professional.
Quality Indicator 8	Percent of missed cases, reported by disorder.

**Table 2 IJNS-07-00049-t002:** NewSTEPs Short-Term Follow-Up national meeting objectives.

**2016 Short-Term Follow-Up National Meeting Objectives**
Provide input and offer expert guidance on challenges in follow-up.
Identify quality improvement initiatives for follow-up.
Develop a toolkit of solutions to common barriers identified.
**2018 Short-Term Follow-Up National Meeting Objectives**
Provide input and offer expert guidance on challenges in follow-up.
Identify focus areas for technical support efforts from the Short-Term Follow-Up Workgroup and NewSTEPs.
Provide an arena in which follow-up staff can network and collaborate on quality improvement efforts.

**Table 3 IJNS-07-00049-t003:** Newborn screening follow-up Quality Indicator data (2018–2020).

Quality Indicator	Year	Number of States Providing Data (*n*)	Median (%)	IQR
4a: Percent of infants that have no recorded final resolution * by 12 months of age with the state newborn screening program following the receipt of an unacceptable dried blood spot specimen.	2018	8	2.32	2.17 (1.89–4.06)
	2019	8	4.06	4.7 (2.31–7.01)
	2020	8	5.41	11.66 (2.06–13.72)
4c: Percent of infants that have no recorded final resolution * by 12 months of age with the state newborn screening program following an out-of-range result from a dried blood spot requiring further clinical diagnostic workup by an appropriate medical professional.	2018	7	1.57	3.48 (1.15–4.63)
	2019	10	1.53	2.12 (0.92–3.04)
	2020	8	2.14	1.64 (1.2–2.84)

* Final resolution for the purposes of this Quality Indicator is defined as a confirmed diagnosis or diagnosis ruled out by an appropriate medical professional.

## Data Availability

The data presented in this study are available on request from the corresponding author. The data are not publicly available due to data security and privacy outlined in memorandums of understanding between APHL and state public health laboratories.

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
