# Peer review of "Establishing a National Community of Practice for Newborn Screening Follow-Up"

_2409-515X, 2021, doi:10.3390/ijns7030049_

Round 1

Reviewer 1 Report

This is a well-written article overall describing the NewSTEPS program - the premise/need, origins, design/set up, components, and activities. The development of programs to connect the newborn screening follow-up community is well laid out. Limitations of establishing long term follow-up are discussed. Although largely descriptive, there is data from states in section 3.3. This seems like an article that will be of interest to the readers of the journal. One critique may be that more data would be useful, if available, about the program.  Section 3.3 could be more robust overall. Table 4 could include IQRs with the median %'s.

Section 3.3 / Table 4 - how many states total were trying to enter data? (paragraph 2 on page 8). Trying to get a sense of how much the data collection fell short from the total # of states trying to currently enter data into the voluntary NewSTEPs Data Repository. 

Any differences for those states with a second NBS? I am curious, as those states have the Primary Care provider actively involved (i.e. sending in the second NBS). Could those states have better communication between state labs and PMDs, more clinical info, and so on.

There are a few small errors in the writing. One is in the Discussion, line 353: "[Error! Bookmark not defined.]." Another error is in line 299: should read "additional states". Line 265: "(whichever comes first)."

It could be helpful to expand on the "limitations that exist in establishing data exchange between the laboratory, follow-up and clinical components of the newborn screening system." (lines 270-271) What are the limitations exactly? How could those be addressed? 

Author Response

How many states total were trying to enter data? 

We have clarified in the text that the number of states and territories that participate in some form in the NewSTEPs Data Repository is 53 (50 states, DC, Puerto Rico and Guam). Not all states have an MOU with NewSTEPs, and not all states have entered Quality Indicator data yet for varying reasons which are further explored in other publications.

Any differences for those states with a second NBS?

This data is not currently available but we will analyze data from one-screen and two-screen states once additional data is entered by states. 

Errors in writing:

The reference on line 353 has been corrected. 
The typo on line 299 has been corrected. 
The punctuation on line 265 has been corrected. 

Limitations:

Data entry is challenging due to limitations that exist in establishing data exchange between the laboratory, follow-up and clinical components of the newborn screening system. This is because in the US each state program is structured differently. Follow-up may be housed separately from laboratory under a different state government department or run by a university or private vendor. Additionally, states use different laboratory information management systems (LIMS) that may not communicate directly.  

Reviewer 2 Report

The authors describe steps towards the formation of a national community of practice for NBS followup programmes in the USA. NewSTEPs have provided a number of important and very useful interventions to facilitate high quality followup throughout the USA. 

Overall the manuscript is well written but is largely descriptive. It would benefit from a more critical evaluation of the success of initiatives efforts to date, as well as outlining plans to build upon and improve these. In particular 

  1. Table 3 - why were the national webinars paused in 2019?  Feedback appeared very positive. Was this due to funding constraints or was it intended as a time-limited project?
  2. The authors describe work on screening follow-up definitions and developing molecular resources. There is no comment about collaboration with key partners such as CLSI and ACMG who have a shared focus on harmonisation and information provision, and may be expected to enhance national efforts towards shared goals.
  3. The proportion of programmes submitting quality indicator data is really very low. Challenges are noted in lines 269-271, and I imagine NewSTEPS has strategies in place to improve voluntary data provision. It would be useful to comment on this.

Author Response

This is very helpful feedback, thank you. The intent of this paper was largely to provide a qualitative overview of the resource center that has been established in support of national NBS follow-up programs. We are collecting additional data in an effort to provide additional, future analyses of state initiatives.

  1. Webinars were previously scheduled quarterly but were paused in 2019 to focus on taskforce projects and again in 2020 due to the Covid-19 pandemic to allow organizers and participants to focus on Covid-19 response. This explanation has been added to the manuscript.  
  2. While we did not work directly with ACMG or CLSI the taskforces drew from their existing resources to develop tools relevant to NBS follow-up staff. This has been added to the manuscript.  
  3. The manuscript has been updated to include a brief mention of the quality improvement program meant to increase engagement with the data repository.  

    NewSTEPs does continue to work with our federal partners to enable states to more efficiently enter a more complete data set. Additional publications on this are expected to be forthcoming as the data is received.  

Reviewer 3 Report

Shortterm and longterm follow-up are essential to maintain and improve the quality of newborn screening programmes. In this paper the authors describe the effort to come to an integrated US- nationwide approach for STFU and LTFU practices.

Although the text itself is well written the whole concept is complicated and not easy to understand, especially for a non-US citizen.

My main observations:

  1. There is no clear definition of short term follow-up and long-term-follow-up. In my opinion, STFU can be regarded mainly as a process quality indicator whereas LTFU is more an effect quality indicator.
  2. The paper highlight in-person meetings “bringing together follow-up personnel” (lines 109-110). It is not clear what kind of professional education these persons have followed: medical, biochemical, nursing etc.. If they differ too much in education and background the results of these meetings will be less than when it is a homogenous group.
  3. When the screening laboratory, under the responsibility of the head of the laboratory indicates that the screening result is not-negative, i.e. actionable, then someone should oversee the follow-up steps in order to ascertain that the infant either will have a second heel prick or it will be referred to a specialist. In my opinion this can be done by someone who does not need to have all the specific knowledge of the condition screened for. Yet, in paragraph 3.4 it is described how such people are educated in interpretation of biochemical newborn screening results (lines 308-310). That seems to be a duplication of the knowledge and activities of the laboratory itself. In addition, it is uncertain to what extent “primary care providers or other medical personnel” (paragraph 3.2) will do with the information received from the STFU-colleague.
  4. Over the past decades the creation of the RUSP has led to a situation that almost all US states are screening for the same conditions which is a large improvement compared to the situation before 2005 or so. States did not want to seen as lagging behind. For the same reason it would certain help if the data on how states are performing in the STFU-metrics could be published (lines 84-95). If a state concludes that it performs worse than other states that may stimulate efforts to improve. Can the authors provide arguments why this is not done yet?

In conclusion: a very interesting paper, with a complicated structure. I would suggest to delete Tables 2 and 3 or move them to a Supplementary file. Both the Introduction and the section on Materials and Methods could be shortened.

Throughout the paper a number of percentages are mentioned: please review the number of decimals to see if they are statistically valid relative to the absolute numbers, e.g. “43.64%” in line 242, which probably should be just “43%”.

Author Response

  1. We agree that the definitions for STFU and LTFU are unclear. This is due to the structure of the US NBS system and differences between state programs. The STFU Workgroup follows the definition provided by APHL in their policy statement on STFU. The LTFU taskforce created a working definition of LTFU that was used to guide the development of the LTFU Landscape Survey. The Short Term Follow-up Workgroup has discussed the possibility of changing their name to NBS Follow-up Workgroup to address the increased attention on LTFU and incorporate all elements of follow-up into their scope of work.  

    APHL does not endeavor to provide a uniform definition, but rather to provide support to programs that autonomously define their own STFU and LTFU parameters. We agree that it would be worthwhile to further explore these nuances in a future publication.  

  2. National meetings are open to all NBS stakeholders and NewSTEPs does not collect information on attendees' educational or professional background. The following has been added to the manuscript: 

    Invitations to these meetings are offered to follow-up program staff from 50 states, DC, Puerto Rico and Guam, with APHL providing travel support for one representative from each US program, however the meetings are open to additional staff and other NBS stakeholders at their own expense.  

  3. Depending on the structure of the state's NBS program, STFU staff may communicate screening results to a child's primary care provider. Often providers have limited knowledge of the disorder and follow-up personnel may explain the interpretation of results so it is important that they understand the basics of molecular genetics to accurately describe the results and nature of the disorder. If treatment is time-sensitive, follow-up personnel may need to be able to explain this to the primary care provider.  
  4. The state specific data is protected by a MOU and is not intended to be shared publicly without the express written approval of each state. De-identified Quality Indicator data is available to states by request.  
  5. Table 3 has been moved to a supplementary section. 
  6. Decimals have been corrected.

Thank you

Round 2

Reviewer 3 Report

Thanks for your amendments